# Remaining Life Assessment for Steel After Low-Cycle Fatigue by Surface Crack Image

**DOI:** 10.3390/ma12050823

**Published:** 2019-03-11

**Authors:** Che-Si Shi, Bin Zeng, Gui-Long Liu, Ke-Shi Zhang

**Affiliations:** Key Lab of Disaster Prevent and Structural Safety, Guangxi Key Lab Disaster Prevent and Engineering Safety, College of Civil Engineering and Architecture, Guangxi University, Nanning 530004, China; shichesi@126.com (C.-S.S.); zengbin379@163.com (B.Z.); guilongliu@163.com (G.-L.L.)

**Keywords:** low-cycle fatigue, pre-cycle, crack distribution, fatigue damage characterization, assessment of remaining life

## Abstract

After pre-fatigue cycles at different strain amplitudes with different *N/N_f_* values (33.3%, 50%, and 75%), specimens of HRB335 steel were subjected to uniaxial tension until failure. By this method the mechanical properties of the specimens after pre-fatigue testing were measured, and the fracture morphology and microscopic morphology in the vicinity of the specimen’s neck surface near the fracture were observed. The verification of the characteristics to be used to estimate the damage caused during the loading cycles was conducted. By observing optical microscope images of the surface area near the neck of the specimens, it was found that the images of surface cracks were significantly different and strongly depended on the number of pre-fatigue cycles the specimen had undergone. In response to this phenomenon, both the microscopic images taken directly from the photos of the surface crack distribution and the binary images based on them were statistically analyzed, and then a parameter, S, denoted as the “unit crack area”, characterizing the cumulative fatigue damage was suggested. Furthermore, the test procedure and the calculation formula for determining the image parameters were summarized, and a method for evaluating the remaining life of steel after low-cycles of reversed tension and compression was proposed.

## 1. Introduction

Fatigue is a common form of failure of engineering structures or materials [1,2,3], which often occurs suddenly under low stress and leads to unforeseen catastrophic accidents. Therefore, researchers have been searching for indicator parameters and methods to reasonably evaluate the remaining life of metals. For example, the accumulated load cycle number has been used to measure a material’s damage (material deterioration), and the concept of fatigue damage was introduced to explain the relationship between cycles under loads and fatigue life [4,5]. In these previous studies, the damage accumulation was assumed to be linear to the number of cycles, and Miner’s rule and the formula for the linear fatigue cumulative damage was applied [6]. In order to reflect the physical process and the corresponding mechanism, researchers tried applying different parameters to measure the damage of metals after cyclic loading from different angles [7,8,9,10,11]. Mesmacque et al. proposed that the degree of strength degradation of a material could be used to measure its damage [12]. Focusing on the mechanism of fatigue damage, many scholars have carried out research on the fatigue life of materials [13,14,15,16,17,18], which is also related to the evaluation of their remaining life. For the damage accumulation of a material, the parameters available to reflect the cumulative damage process and whether they can be directly measured is a key issue. A common method is to identify damage parameters by referring to experimental observations. For example, Corten-Dolan considered that fatigue damage originates from micro-crack accumulation, and so suggested that [19]:(1)$$D=mrN^{\mu }$$
where m is the number of crack cores; r is the crack propagation coefficient; N is the number of cycles; and μ is the constant to be determined associated with the stress level. Considering that the cracks gradually propagated in the material, Manson and Halford suggested that the damage accumulation could be expressed as [20]:(2)D=1af[a0+(af−a0)(NNf)23Nf0.4]
where $a_{f}$ and $a_{0}$ are respectively the final and initial crack lengths of the material and $N_{f}$ is the fatigue life (number of cycles) related to specified cycle loading. Chaboche gave a description of the fatigue damage accumulation law based on continuous damage mechanics [21]:(3)D=1−[1−(NNf)11−α]11+β
where α is a parameter related to the loading maximum stress and average stress and β is a material constant.

Since complex processes are involved, it is actually difficult to measure damage in experiments using the above damage definitions. These definitions do not directly reflect the evolution of the microstructure of materials and it is difficult to describe the actual damage process.

After having undergone cyclic loading, the microstructures in a material will inevitably be changed. This leads to the request for reasonable characterization parameters to describe the relation of this change to the damage of materials experiencing plastic deformation [22,23,24,25,26,27,28,29]. Only with these parameters can one identify the evolution of microstructures during cycling through actual measurements. An experimental study [30] found that, although having undergone tensile and compressive cycles at specified different strain amplitudes, the differences in the morphology of the surfaces of Q235 steel specimens are difficult to distinguish. However, if the specimens are subjected to large deformed stretching, the surfaces of the specimens will become rough and densely distributed with tiny cracks; the more cycles, the more obvious the distribution of cracks. The “cracks” are actually micro-ruptures on the shallow surface of the specimen caused by large deformed stretching, which can be regarded as the revealing of accumulated distributed fatigue damage. It seems that the fatigue damage caused by the low-cycle fatigue cycle can be presented on the specimen’s surface using this method. To further explore and verify this method, this paper carried out the following studies: using HRB335 steel as study object, the specimens were subjected to a number of cycles under specified strain amplitudes which increased tension until they fractured, and then the distribution of cracks on the adjacent surface of the neck of specimens were photographed and recorded with an optical microscope circumferentially. The binary images of the distributed cracks were extracted and the distribution crack areas were statistically analyzed. Through comparing the fatigue life through testing, the relationship between the statistical value of the distribution crack area and the remaining life of the material after the pre-fatigue cycle could be studied and verified. Based on this, a new quantitative description method for fatigue cumulative damage of materials is proposed.

## 2. Materials, Samples and Tests

The low alloy hot rolled steel HRB335 was used as the test material. Its main chemical composition is shown in Table 1, and its mechanical properties are shown in Table 2. The values listed in Table 1 and Table 2 are nominal values provided by the manufacturer (Liuzhou Iron and Steel Company, LTD., Liuzhou, China), which comply with Chinese national standards (GB/T3274-2007) [31]. The specimen was a smooth round bar; its dimensions are shown in Figure 1. After cutting and polishing, the surface roughness of the working section of the specimen was Ra 0.4.

The tensile and compressive cycle test was carried out at room temperature by a MTS809 fatigue testing machine. The experiment was controlled by a constant strain amplitude sine wave with frequency of 1 Hz. The relative total strain rate variation range of this paper was 0.012–0.04 s^−1^ (with respect to the strain amplitude range 0.003–0.01). Both comply with the regulations of ASTM-E468-90 (2004). The mathematical expectation of the fatigue life of the material measured at the specified strain amplitude, $\varepsilon_{a}$, was denoted as $N_{f}$, and the fatigue tests with the strain amplitudes of 0.003, 0.004, 0.005, 0.008, and 0.01 were conducted. At least 3 specimens were used in each strain amplitude test, and the fatigue life value of an individual specimen was denoted as $N_{f}^{i}$, and its average was ${\bar{N}}_{f}$, see Table 3. Then, referring to Table 3, three pre-fatigue loading conditions ($N/{\bar{N}}_{f}$ values of 33.3%, 50%, and 75%, respectively, N is the number of cycles) were conducted for five strain amplitudes, as shown in Table 4. All steady hysteresis loops at various strain amplitudes in the tests are shown in Figure 2. During cycles, there was no obvious cyclic hardening and softening.

## 3. Damage Identification and Testing of Specimens Subjected to Different Fatigue Cycles

The macroscopic deformation of the specimens subjected to the fatigue cycle was regarded as homogeneous. Due to the randomness and complexity of the material structure, the internal deformation and accumulated distribution damage was actually uneven. In order to identify the damage of specimens that experienced different pre-fatigue testing, the following experimental analyses were performed.

### 3.1. Changes to the Mechanical Properties of Materials under Uniaxial Tension after Pre-Fatigue Testing

Figure 3 shows the nominal stress–strain curves for specimens uniaxially stretched to failure after undergoing different life consumptions ($N/{\bar{N}}_{f}$) cyclic loading at a strain amplitude of 0.005. It can be seen from this figure that the nominal stress–strain curves only for the specimen directly stretched without pre-circulation have a clear yielding platform; all other curves are similar, without obvious difference.

The yield strength, tensile strength, and working section (40 mm) elongation determined are plotted in curves in Figure 4. The figure shows that the yield strength of the material, denoted by $\sigma_{s}$, with the increase of $N/{\bar{N}}_{f}$, does not change much but disperses significantly. This indicator does not seem like an easy method for evaluating the remaining life of the specimens. The test of the residual strength of the specimens subjected to pre-fatigue at different strain amplitudes with different consumed relative life values (equivalent to the number of pre-fatigue cycles performed) was carried out by placing them under tension until they fractured, and the residual strength was obtained from the record of tested maximum loads and is expressed in the figure by $\sigma_{\max}$. The results with respect to different relative life consumption (Figure 4a) were similar to that of Q235 steel and austenitic stainless steel [30,32]. Some researchers believe that the accumulation of fatigue damage in materials will reduce the tensile strength of the material, and attempt to establish a relationship between the remaining life and residual strength on this basis [33]. But the results of Figure 4a show that the increase in the accumulation of damage experienced by the material does not necessarily lead to a decrease in the residual strength, and the remaining life of the material cannot be inferred from the residual strength. Let us examine the ratio of elongation (residual ductility) of the working section (with an initial length of 40 mm) after pre-fatigue indicated by A_40_. It can be seen from Figure 4b that the elongation of the specimens after the cycle had a tendency to decrease as the number of pre-cycles $N/{\bar{N}}_{f}$ increased. However, the range of variation was fairly small, so it was not suitable to be an indicator for assessing the remaining life of the specimens.

### 3.2. Differences in Fracture Morphology of Specimens Tension-Fractured after Pre-Fatigue

Figure 5 shows the fracture morphology of uniaxially tension fractured specimens after $N/{\bar{N}}_{f}$ = 0%, 33.3%, 50%, and 75% pre-cycles at a 0.005 constant strain amplitude. It can be seen that when $N/{\bar{N}}_{f}$ ≤ 75%, the fracture morphology exhibited the characteristics of dimple-type ductile fractures, which were similar to the ones found by direct uniaxial tension without pre-fatigue. This means that it is difficult to find an indicator parameter for determining the remaining life from the fracture topography.

### 3.3. Damage Information about the Neck Surface of the Pre-Cycle Specimens after Tension Fracture

After having undergone different pre-fatigue cycles, the specimen surface under the optical microscope was still smooth, and it was difficult to observe the difference caused by different fatigue cycles. However, after uniaxially stretching to break, the difference in crack distribution associated with the number of cycles could be observed by the naked eye on the surface of the neck portion of the specimens. Figure 6 shows the appearance of specimens subjected to different pre-fatigue cycles at a constant strain amplitude of 0.005. The test results at other strain amplitudes are not shown due to page limitations, but they are similar. Figure 6a shows the smooth surface of the original specimen. Figure 6b shows the specimen which was directly subjected to uniaxial tension until fracture without pre-fatigue, and the surface was rough compared to the original, but no crack was found on the surface. Figure 6c–e are specimens with pre-cycle $N/{\bar{N}}_{f}$ of 33.3%, 50%, and 75%, respectively. They had obvious rough bands and micro-cracks on the neck surface, and the larger the number of fatigue cycles, the more obvious the rough bands and micro-cracks. As mentioned earlier (Section 1), micro-cracks are the micro-rupture appearance on the shallow surface of the specimen caused by large deformation stretching, which reveals accumulated distributed fatigue damage. This is most likely related to some initial processing of micro-structures on the surface or subsurface of the material. During cyclic loading, the damage gradually evolves on these structures and deepens. At last, it is revealed by large deformation stretching. Figure 6f is the specimen which was fractured by fatigue, and the surface still looked smooth. These results were similar to those of the Q235 steel test [30].

The surface images of the specimens were further observed by using an optical scanning microscope. The 100× lens was selected and the following photographing parameters were set: the contrast ratio was 42, edge enhancement was 2.1, and brightness was medium. In order to avoid the differences caused by the photographic field of view, the photos were taken from a uniform distance of 2 mm from the fracture and the specimen was rotated in a circle (360 degrees, 36 degrees per 1 frame). Figure 7a is a schematic diagram of the photographic sequence. Figure 7b–k are photomicrographs (the vertical is the annular direction of the specimen) of the surface at the neck area adjacent fracture of the specimen after the consumed life of $N/{\bar{N}}_{f}$ = 75% of pre-fatigue. 

This figure shows the apparently rough and distributed tiny cracks on the surface of the fatigue damaged specimen after uniaxial tension. It should be noted that these were not observed when specimens under direct uniaxial tension broke without pre-fatigue or were directly fatigued to fracture.

#### 3.3.1. Image Processing and Analysis

For further quantitative analysis, image processing of the original photo taken with a microscope was conducted by using the Matlab (version: R2010a) image analysis program to extract the rough strip and micro-crack feature information of the specimens. In this paper, the image characteristics of the specimen surface were processed as follows: original image → image preprocessing → threshold preference → binarization → image morphology processing → obtaining the statistical “crack area” parameter. See Table 5 for a description of the methods (functions) used throughout this process.

After the different cycles, the specimens were uniaxially stretched and then microscopic images of the surfaces in the area adjacent to the fractures were taken, and image features were extracted. Only the surface images and corresponding binary images of a 36 degree range of specimens that had undergone the cycles with a strain amplitude of 0.005 and different relative life $N/{\bar{N}}_{f}$ values are shown here, as shown in Figure 8. The black area of the processed binary image is regarded as the “crack”, and the ratio of the “crack” area to the area of the view area is defined as the unit “crack area”, and is denoted as S.
(4)$$S=S_{c}/S_{t}$$
where $S_{c}$ is the area of the crack determined from the binary image, and $S_{t}$ is the total area of the view. S is the dimensionless quantity directly reflecting the change of material microstructure and damage accumulation caused by the fatigue cycle.

Figure 9 shows the variation of the parameter S with the increase of $N/{\bar{N}}_{f}$ at different strain amplitudes. In the figure, the black hollow dots represent the individual test values for different views in a circle, and the red solid dots represent their statistical mean. They show that each mean curve increases monotonically with $N/{\bar{N}}_{f}$, which reflects the irreversibility of damage accumulation with cyclic loading. The dispersion of the S value became larger with an increase in consumed life, mainly because the fatigue damage distribution was not uniform, and it increased with the pre-cycle number.

#### 3.3.2. Influence of Photographic Parameters

The above results are obtained with the selected photographic parameters selected of acontrast ratio of 42, edge enhancement of 2.1, and medium brightness. In order to understand the influence of these three photographic parameters on the image processing results, the following comparisons were made on each parameter variation (the other parameters were unchanged): (1) The contrast ratios were 42, 50, 100, and 150, the original image and the corresponding binary image are shown in Figure 10; (2) the edge enhancement parameters were 2.1, 5, 10, and 20, the original image and the corresponding results are compared in Figure 11; (3) the brightness is darker, medium, and brighter, and the results are shown in Figure 12. Figure 13 shows the effect of these parameter selections on the numerical results of image processing. It can be seen from the figure that the difference in contrast ratio had little effect on the data of parameter S, see Figure 13a. The edge enhancement parameters had a great influence on the data results, see Figure 13b. The brightness change from darker to medium had a greater impact on the results, while the effect of the change from medium to brighter was less, see Figure 13c. These results show that in order to reduce the dispersion of the resulting data and human intervention, uniform imaging parameters (especially edge enhancement parameters) are required for image processing.

## 4. The Remaining Life of Material Fatigue and Predictions

The above experimental observations show that the parameter S has a quantitative relationship with the cycle consumption life, which can reflect the deterioration related to irreversible fatigue damage of the material. For the convenience of comparison, Table 6 lists the S mean values of the parameters measured for different strain amplitudes and consumed life $N/{\bar{N}}_{f}$.

Because the parameter S measured in the experiments with the respective strain amplitudes were numerically similar to the increase of $N/{\bar{N}}_{f}$, the relationship between the parameters S and $N/{\bar{N}}_{f}$ can be approximated by Equation (5) (as mentioned earlier, $N_{f}$ is the mathematical expectation of ${\bar{N}}_{f}$). In this equation the damage parameter D is defined, which takes the value within the interval of (0,1); 0 corresponds to the original state without damage and 1 corresponds to fatigue failure.
(5)S=A+B⋅(NNf)α, D=(S−A)/B
where, A, B and α are constants to be determined. They can be obtained by fitting the test data listed in Table 6. Table 7 shows the constants A, B and α obtained by the test results under different strain amplitude cycles according to Equation (5).

It can be seen from Figure 9 that the S–$N/N_{f}$ curves approximately reflect the law of damage evolution, although when measured in the tests with different strain amplitudes they had a few differences, see the red dashed curves. Therefore, the S–$N/N_{f}$ data could be measured from the cyclic tests at a strain amplitude to calibrate the parameters of Equation (5), and the parameters could be used to predict the remaining life in other experiments with different strain amplitudes. When the pair of data S and cycle number $N/N_{f}$ was measured at other strain amplitudes, the relative life estimation for $N_{f}$ could be given by Equation (5), that is $N_{f}=N/{\left[\left(S-A\right)/B\right]}^{1/\alpha }$, and then the remaining life prediction could also be given by $N_{f}-N$. To verify the rationality and validity of the method, one can take the predicted value and compare it with the measured value.

In Figure 9, the curves measured at the strain amplitudes of 0.01 and 0.004 were respectively the highest and lowest of all curves, that is, their deviation from the mean was the largest. If these two curves are used to calibrate the parameters of Equation (5) separately, the error of the predictions mentioned above will be the largest. Therefore, if the largest error of the prediction result is still within the allowable range, it indicates that the method is reasonable and effective. Table 8 lists the number of cycles *N* of the three sets of cycle tests under different strain amplitudes, and includes *N*′, *N*″, and *N*‴ which correspond to the different numbers of cycles, the corresponding value of parameter S ( *S*′, *S*″, and *S*‴), and the relative life predictions *N*′*_f_*, *N*″*_f_*, and *N*‴*_f_* obtained by the parameters of Equation (5) calibrated at strain amplitudes of 0.01 and 0.004, and the remaining life prediction $N_{f}^{\prime }-N^{\prime }$,$N_{f}^{\prime \prime }-N^{\prime \prime }$, $N_{f}^{\prime \prime \prime }-N^{\prime \prime \prime }$, and the mean prediction, ${\tilde{N}}_{f}$, for *N*′*_f_*, *N*″*_f_*, and *N*‴*_f_*. The error comparison between the predicted value and the corresponding measured value is showed in Figure 14. The abscissa of the graph is the measured fatigue life, and the ordinate is the predicted life; the thick solid line in the figure represents the ideal prediction, and the area with dotted line on both sides represents the interval of a factor of 2, and the area with thin solid lines on both sides represents the interval of a factor of 3. The solid triangles pointing downwards and pointing upwards are the predictions given by Equation (5) for the calibration of the measured data under $\varepsilon_{a}$ = 0.004 (Case 1) and $\varepsilon_{a}$ = 0.01 (Case 2), respectively. It can be seen from Figure 14a that the predicted life was determined to be within the reasonable double factor region compared to the actual measurements, regardless of whether the parameters of Equation (5) were determined under Case 1 or Case 2. As the number of pre-cycles increased, the fatigue life predicted value was closer to the measured value. From Figure 14b and Table 8, it can be known that the parameters of Equation (5) were determined under Case 2 to predict the remaining fatigue life for pre-fatigue at a strain amplitude of 0.004, or the parameters were determined under Case 1 to predict the same at a strain amplitude of 0.01. The predicted remaining fatigue life had an error rate exceeding twice the actual measurement, but was within a factor of 3. For cycles at other amplitudes, whether the parameters were determined under Case 1 or Case 2, the predictions of remaining fatigue life were still within the reasonable area of a factor of 2. Considering that the prediction was made according to the curve with the maximum deviation and the fatigue tests with dispersion, the rationality and validity of the results are acceptable. It should be pointed out that: (1) the measured remaining life in Figure 14b refers to the value using the measured mean cycle life of Table 3 minus the measured pre-fatigue cycles; and (2) the predictions give an overestimation of fatigue life and remaining life in Case 2 and an underestimation in Case 1.

The forecast data *N’_f_*, *N’’_f,_* and *N’’’_f_* for $N_{f}$ presented in Figure 14 are also listed in Table 8. The relationship between strain amplitude $\varepsilon_{a}$ and predicted life $N_{f}$ can be seen in Figure 15. From Figure 15, the lower and upper boundary of the estimations for the predicted lifetimes of the tests at different strain amplitudes, in which the parameters of Equation (15) are identified under the conditions Case 1 and Case 2, can be observed. Based on the measured data in the figure, the rationality of the method for predicting the low-cycle fatigue life of the material can be preliminarily confirmed.

## 5. Conclusions

In this paper, the uniaxial stretching to fracture of specimens that were subjected to pre-fatigue testing with different cycles at different strain amplitudes was conducted in order to explore the test method for predicting the remaining life of the material. The following conclusions were obtained:(1)Material damage caused by fatigue cycles will lead to tiny cracks appearing on the surface of the specimen during uniaxial stretching.(2)The “unit crack area”, S, or the distribution density of the tiny crack distribution, on the surface of the specimen neck is related to the pre-cycle number of the specimen subjected to pre-fatigue.(3)The damage of metal during low-cycle fatigue can be characterized by the parameter, S, to reflect the microstructure evolution of the material, which increases with the fatigue cycle number, and can be used to predict the fatigue life and remaining life of materials.

## Figures and Tables

**Figure 1 materials-12-00823-f001:**
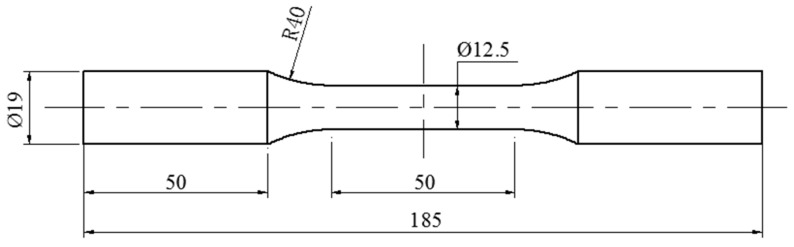
The smooth round bar specimen (unit: mm).

**Figure 2 materials-12-00823-f002:**
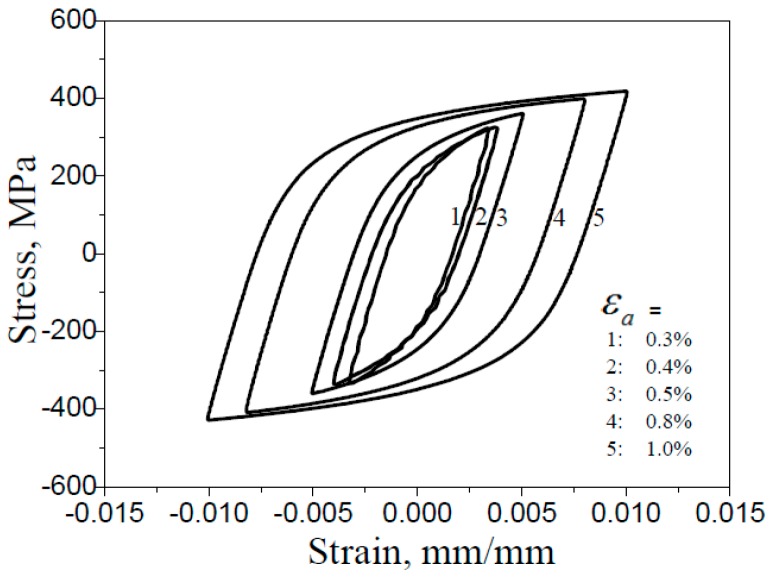
Steady hysteresis loops at various strain amplitudes by test.

**Figure 3 materials-12-00823-f003:**
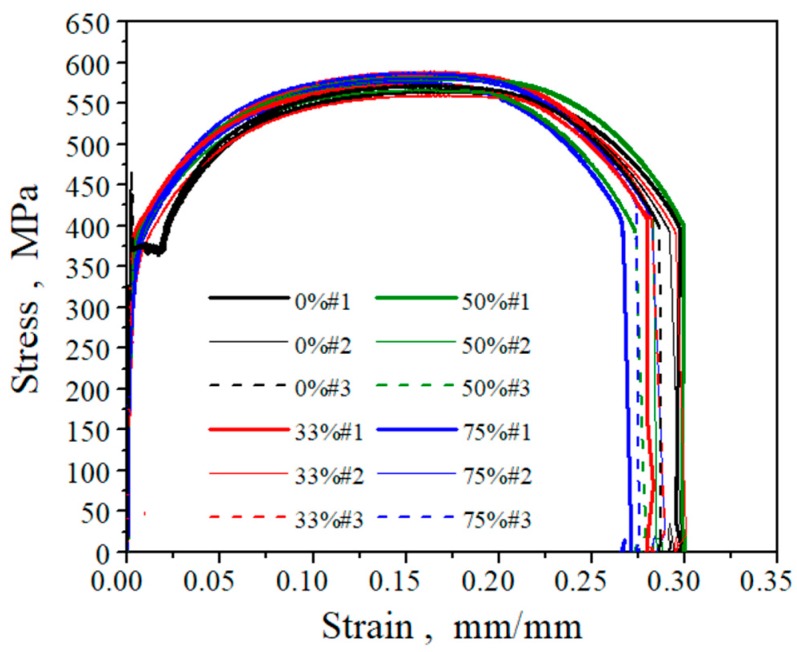
The test nominal stress–strain curves of the material under uniaxial tension.

**Figure 4 materials-12-00823-f004:**
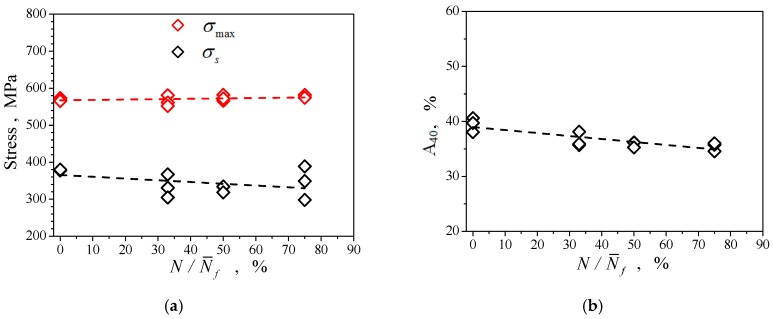
The uniaxial tensile properties of the consumed life of specimens $N/{\bar{N}}_{f}$. (**a**) Residual strength and yielding strength; (**b**) Ratio of elongation.

**Figure 5 materials-12-00823-f005:**
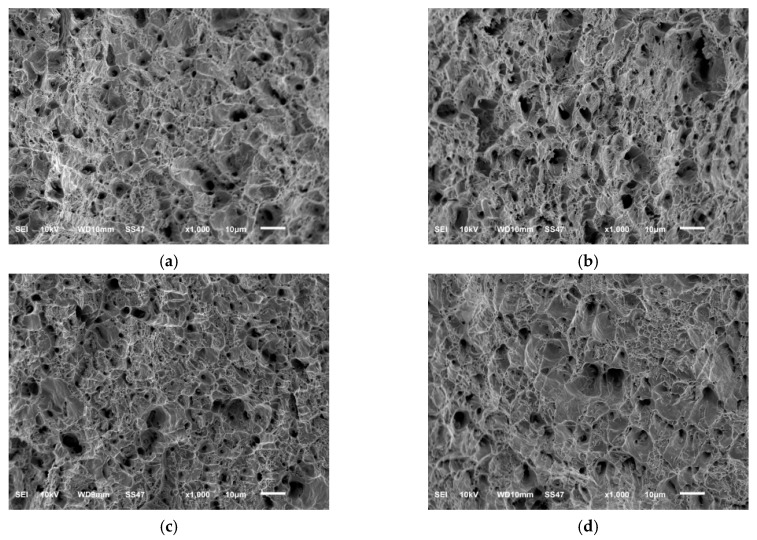
Scanning electron microscope (SEM) fracture images of specimens fractured by uniaxial tension after pre-cycles. (**a**) $N/{\bar{N}}_{f}$ = 0%; (**b**) $N/{\bar{N}}_{f}$ = 33.3%; (**c**) $N/{\bar{N}}_{f}$ = 50%; (**d**) $N/{\bar{N}}_{f}$ = 75%.

**Figure 6 materials-12-00823-f006:**
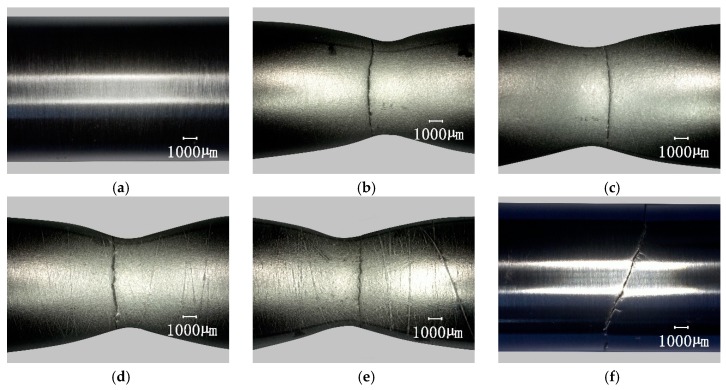
Fractured surface morphology of pre-cycle specimens by uniaxial tension. (**a**) Original specimen; (**b**) $N/{\bar{N}}_{f}$ = 0%; (**c**)$N/{\bar{N}}_{f}$ = 33.3%; (**d**) $N/{\bar{N}}_{f}$ = 50%; (**e**) $N/{\bar{N}}_{f}$ = 75%; (**f**) $N/{\bar{N}}_{f}$ = 100%.

**Figure 7 materials-12-00823-f007:**
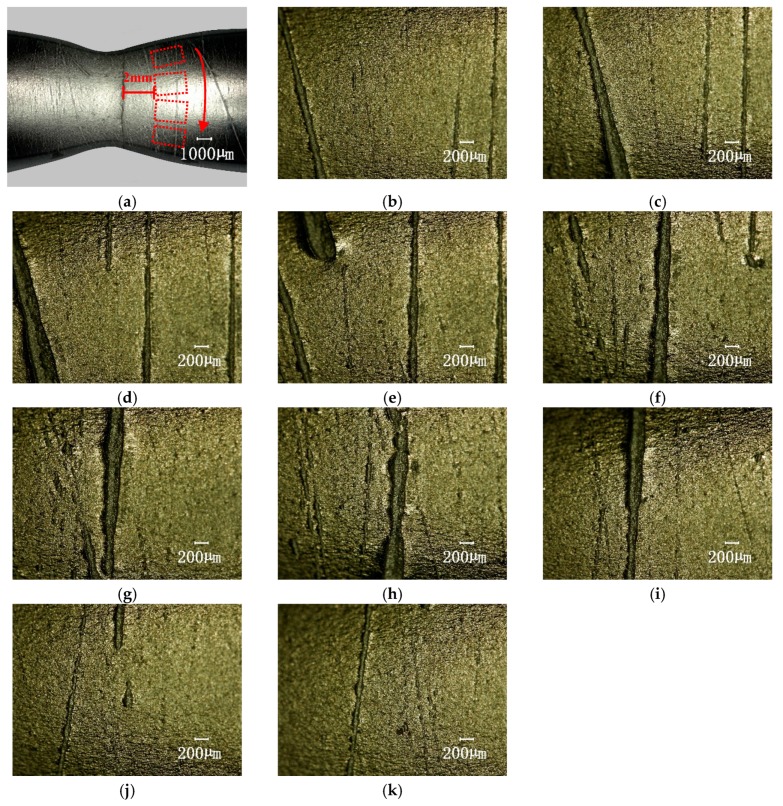
The surface topography at the neck area adjacent fracturing of the specimen by uniaxial tension after the consumed life of $N/{\bar{N}}_{f}$ = 75% of the pre-fatigue value (100× magnification). (**a**) Shows the photographic sequence; (**b**–**k**) show 10 photos of the surface topography, 36 degrees per 1 frame.

**Figure 8 materials-12-00823-f008:**
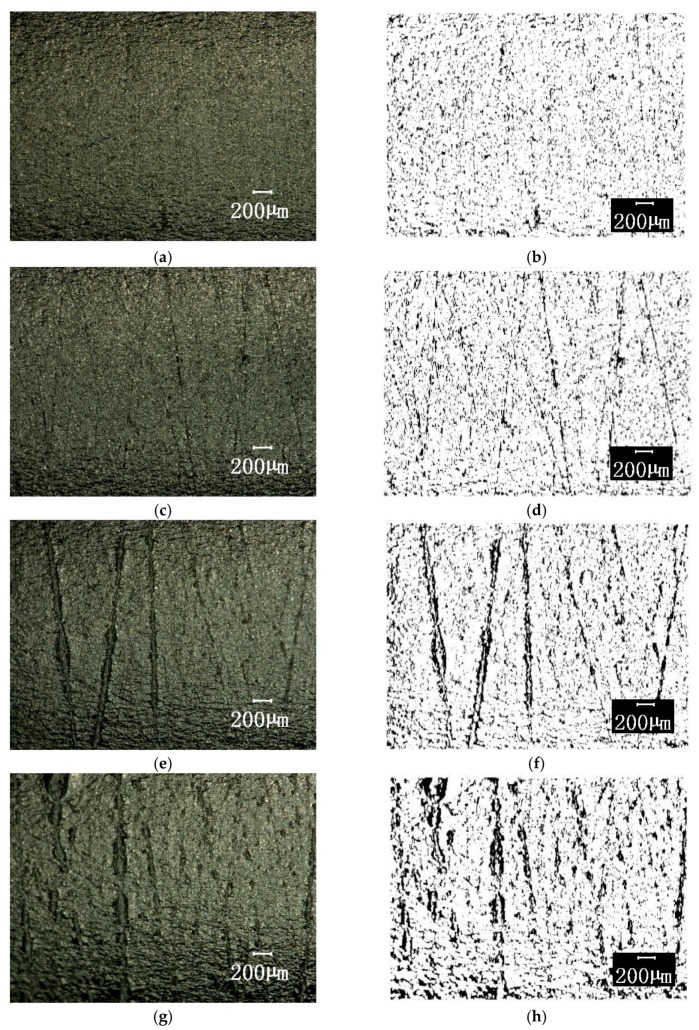
The surface topography of the area adjacent to the fractures of the specimens after pre-fatigue at a strain amplitude 0.005 and its binary image. (**a**) $N/{\bar{N}}_{f}$ = 0%, original image; (**b**) $N/{\bar{N}}_{f}$ = 0%, binary image; (**c**) $N/{\bar{N}}_{f}$ = 33.3%, original image; (**d**) $N/{\bar{N}}_{f}$ = 33.3%, binary image; (**e**) $N/{\bar{N}}_{f}$ = 50%, original image; (**f**) $N/{\bar{N}}_{f}$ = 50%, binary image; (**g**) $N/{\bar{N}}_{f}$ = 75%, original image; (**h**) $N/{\bar{N}}_{f}$ = 75%, binary image.

**Figure 9 materials-12-00823-f009:**
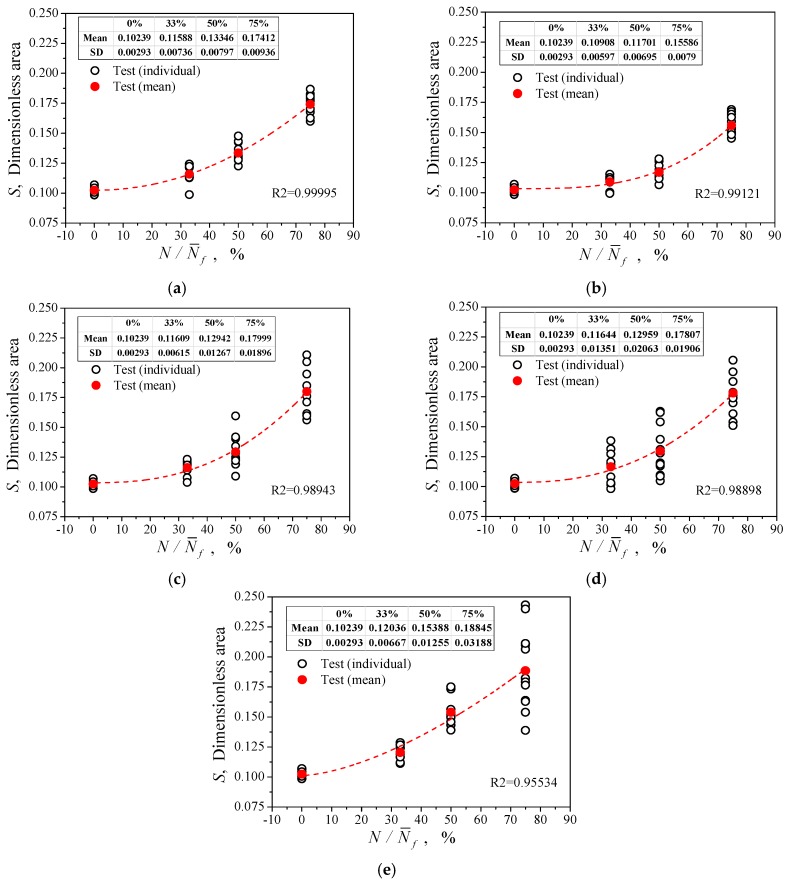
Relationship between the parameter S, and consumed life after pre-fatigue by uniaxial tension to fracture. (**a**) $\varepsilon_{a}$ = 0.003; (**b**) $\varepsilon_{a}$ = 0.004; (**c**) $\varepsilon_{a}$ = 0.005; (**d**) $\varepsilon_{a}$ = 0.008; (**e**) $\varepsilon_{a}$ = 0.001.

**Figure 10 materials-12-00823-f010:**
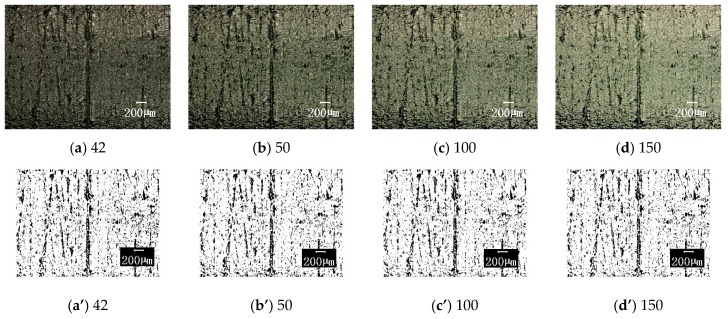
The surface topographies under different contrasts with corresponding binary images (100× magnification). The contrast ratios are respectively taken as: (**a**) and (**a**’) 42; (**b**) and (**b**’) 50; (**c**) and (**c**’) 100 and (**d**) and (**d**’) 150; for original image and the corresponding binary image, respectively.

**Figure 11 materials-12-00823-f011:**
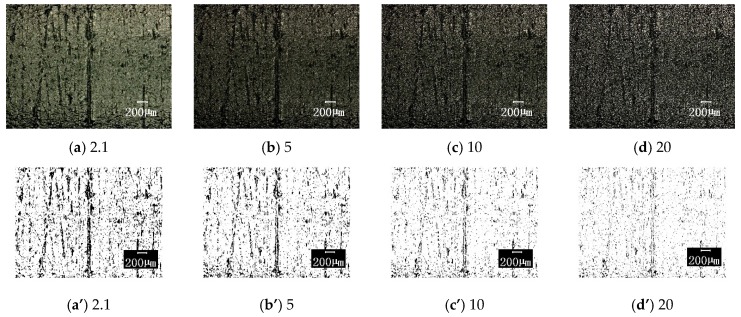
The surface topographies under different edge-enhancements with corresponding binary images (100× magnification). The parameters of edge-enhancement are respectively taken as: (**a**) and (**a**’) 2.1; (**b**) and (**b**’) 5; (**c**) and (**c**’) 10 and (**d**) and (**d**’) 20; for original image and the corresponding binary image, respectively.

**Figure 12 materials-12-00823-f012:**
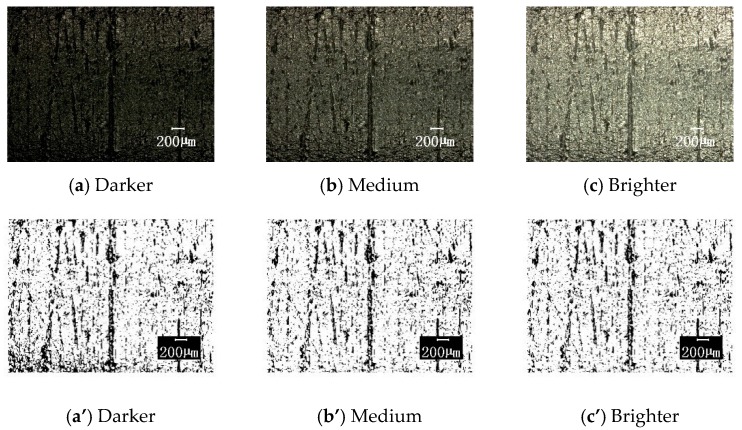
The surface topographies under different brightness levels with corresponding binary images (100× magnification). The parameters of brightness level are respectively taken as: (**a**) and (**a**’) Darker; (**b**) and (**b**’) Medium and (**c**) and (**c**’) Brighter; for original image and the corresponding binary image, respectively.

**Figure 13 materials-12-00823-f013:**
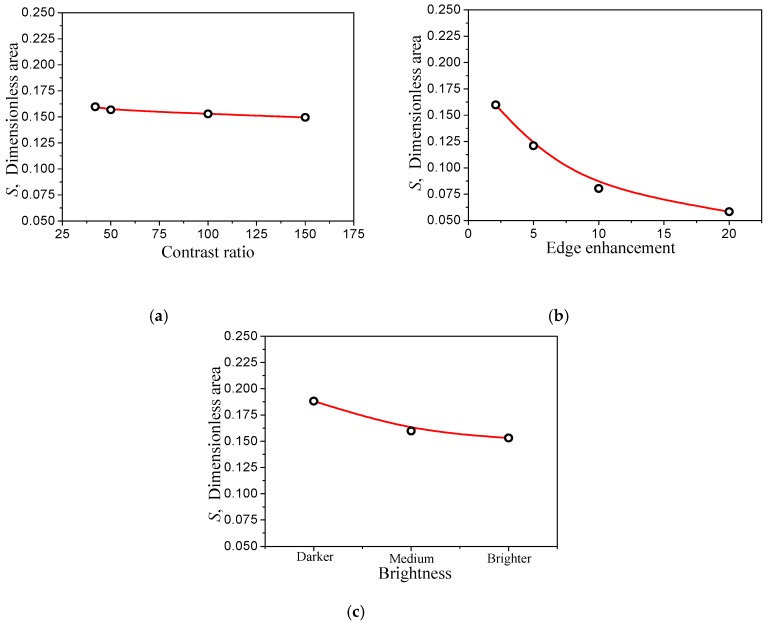
Influence of different camera parameters on the image processing result data. Comparisons respectively by (**a**) contrast ratio; (**b**) edge enhancement and (**c**) brightness.

**Figure 14 materials-12-00823-f014:**
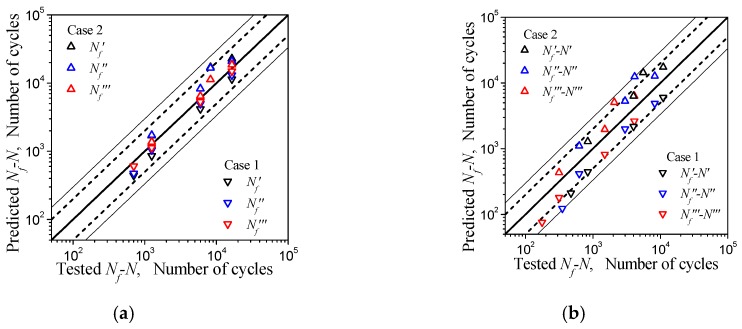
Comparison between predictions and tests. Error assessment of predictions: (**a**) for fatigue life; (**b**) for remaining life.

**Figure 15 materials-12-00823-f015:**
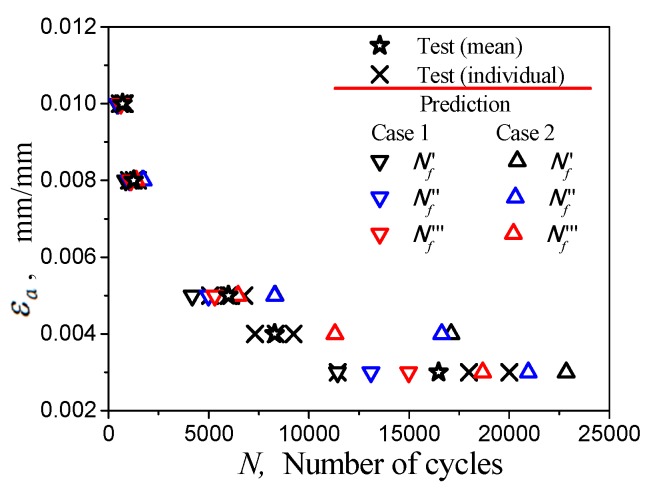
Prediction of fatigue life curves for fatigue cycles at different strain amplitudes.

**Table 1 materials-12-00823-t001:** Chemical composition of HRB335 (%).

C	Si	Mn	P	S	As	V
0.21	0.48	1.16	0.022	0.023	0.025	0.006

**Table 2 materials-12-00823-t002:** Mechanical properties of HRB335 steel.

E/MPa	σ_s_/MPa	σ_b_/MPa
210	355	520

**Table 3 materials-12-00823-t003:** The fatigue life of HRB335 steel at different strain amplitudes.

$\varepsilon_{a}$	$N_{f}^{i}$	${\bar{N}}_{f}$
0.003	11435/20003/17991	16,476
0.004	9233/7313/8345	8297
0.005	6784/5993/5991/6122/5066	5991
0.008	1025/1266/1472	1254
0.010	563/735/813	704

**Table 4 materials-12-00823-t004:** The pre-fatigue cycle number.

$\varepsilon_{a}$	$N_{}^{}$ $\left(N/{\bar{N}}_{f}\;=\;33.3\%\right)$	$N_{}^{}$ $\left(N/{\bar{N}}_{f}\;=\;50\%\right)$	$N_{}^{}$ $\left(N/{\bar{N}}_{f}\;=\;75\%\right)$
0.003	5437	8238	12,357
0.004	2738	4149	6222
0.005	1997	2995	4493
0.008	414	627	940
0.010	232	352	528

**Table 5 materials-12-00823-t005:** Process of image processing by applying Matlab.

Step	Function	Parameter	Features
1	rgb2gray	-	Convert the original image to a grayscale image
2	adapthisteq	-	Equalize the gray histogram to increase image contrast; suppress secondary information in the image and enhance important information
3	graythresh	-	Threshold optimization using the maximum interclass variance method
4	im2bw (I, level)	I is the equalized grayscale image, and level is the threshold determined by step 3 (value range (0, 1))	Binarization
5	Strel (‘line’, LEN, DEG)	LEN takes 11, DEG takes 90	Image morphology processing: extract the rough shape of the test piece and the main shape features of the micro-crack, and weaken the secondary image information
6	Imdilate (BW, se)	BW is the binary graph obtained by step 4, and se is the structural element returned by the step 5 Strel function.	
7	bwarea	-	Statistical surface “crack area”

**Table 6 materials-12-00823-t006:** The mean values with the corresponding parameters *S* measured under different strain amplitudes and consumed life $N/{\bar{N}}_{f}$.

$\varepsilon_{a}$	${\bar{N}}_{f}$	S			
		$N/{\bar{N}}_{f}\;=\;0\%$	$N/{\bar{N}}_{f}\;=\;33.3\%$	$N/{\bar{N}}_{f}\;=\;50\%$	$N/{\bar{N}}_{f}\;=\;75\%$
0.003	16,476	0.10239	0.11588	0.13346	0.17412
0.004	8297	0.10239	0.10908	0.11701	0.15586
0.005	5991	0.10239	0.11609	0.12942	0.17999
0.008	1254	0.10239	0.11644	0.12959	0.17807
0.010	704	0.10239	0.12036	0.15388	0.18845

**Table 7 materials-12-00823-t007:** Fitting results of fatigue accumulation formula parameters.

$\varepsilon_{a}$	*A*	*B*	*α*
0.003	0.10244	0.12936	2.05306
0.004	0.10334	0.12974	3.14702
0.005	0.10349	0.15544	2.48041
0.008	0.10343	0.14805	2.39624
0.010	0.10117	0.13933	1.56606

**Table 8 materials-12-00823-t008:** Predictions of life and remaining life at different strain amplitudes (the parameters of Equation (5) calibrated from the data measured at strain amplitudes of 0.004 and 0.010).

**From the Data Measured at Strain Amplitude 0.004**													
$\varepsilon_{a}$	$N_{}^{\prime }$	$S_{}^{\prime }$	$N_{f}^{\prime }$	$N_{f}^{\prime }-N^{\prime }$	$N_{}^{\prime \prime }$	$S_{}^{\prime \prime }$	$N_{f}^{\prime \prime }$	$N_{f}^{\prime \prime }-N^{\prime \prime }$	$N_{f}^{\prime \prime \prime }-N^{\prime \prime \prime }$	$S_{}^{\prime \prime \prime }$	$N_{f}^{\prime \prime \prime }$	$N_{f}^{\prime \prime \prime }-N^{\prime \prime \prime }$	${\tilde{N}}_{f}$
0.003	5437	0.11588	11,424	5987	8238	0.13346	13,102	4864	12,357	0.17412	14,981	2624	13,168
0.005	1997	0.11609	4174	2177	2995	0.12942	4987	1992	4493	0.17999	5311	818	4823
0.008	414	0.11644	858	444	627	0.12959	1042	415	940	0.17807	1120	180	1006
0.010	232	0.12036	442	210	352	0.15388	475	123	528	0.18845	604	76	506
**From the Data Measured at Strain Amplitude 0.010**													
$\varepsilon_{a}$	$N_{}^{\prime }$	$S_{}^{\prime }$	$N_{f}^{\prime }$	$N_{f}^{\prime }-N^{\prime }$	$N_{}^{\prime \prime }$	$S_{}^{\prime \prime }$	$N_{f}^{\prime \prime }$	$N_{f}^{\prime \prime }-N^{\prime \prime }$	$N_{f}^{\prime \prime \prime }-N^{\prime \prime \prime }$	$S_{}^{\prime \prime \prime }$	$N_{f}^{\prime \prime \prime }$	$N_{f}^{\prime \prime \prime }-N^{\prime \prime \prime }$	${\tilde{N}}_{f}$
0.003	5437	0.11588	22,848	17,411	8238	0.13346	20,955	12,717	12,357	0.17412	18,679	6322	20,827
0.004	2738	0.10908	17,099	14,361	4149	0.11701	16,631	12,482	6222	0.15586	11,305	5083	15,011
0.005	1997	0.11609	8317	6320	2995	0.12942	8297	5302	4493	0.17999	6464	1971	7692
0.008	414	0.11644	1699	1285	627	0.12959	1730	1103	940	0.17807	1373	434	1600

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
