# Peer review of "Remaining Life Assessment for Steel After Low-Cycle Fatigue by Surface Crack Image"

_materials, 2019, doi:10.3390/ma12050823_

Round 1

Reviewer 1 Report

The submitted manuscript entitled ‘Remaining Life Assessment for Steel after Low-Cycle Fatigue by Surface Crack Image’ deals with the life-time prediction in the case of low-cycle fatigue based on the surface crack analyisis of tensiled samples. In the Introduction, the Authors state: ‘Since involving complex processes, it is actually difficult to measure damage in experiments by using above damage definitions. Because they do not directly reflect the evolution of microstructure of materials and it is difficult to describe the actual damage process.’ – according to this, this Reviewer could not find any progress in the paper, the proposed process seems to be quite cumbersome; however, interesting. During the careful review of the manuscript, the following issues arose.

- Please provide an official e-mail address (instead of commercial) for all the Authors, but at least for the corresponding Author.

- Are the chemical composition and mechanical values listed in tables 1 and 2 nominal values or measured? If the latter, how were they measured?

- In fig 1 the radius between the head and the measured part of the sample is missing.

- How was the damage levels (33.3%, 50% and 75%) selected?

- Fig 2 could be omitted.

- In its present form fig 3 is too messy, the curves are overlaping please reconsider.

- In fig 4 not the yield, but the maximum stress is increasing, presumably because of the presence of the crack, working as a stress concentrator.

- In fig 6 the scalebars are almost invisible, while in the micrograph subfid in fig 7 they are completely missing as well as in figs 8, 10, 11 and 12.

- The value of S is surely distorted by the features of the surface (roughness) and in the opinion of this Reviewer cannot be related to the amount of the existing cracks.

- Please give the equations of the fitted curve in fig 9 as well as their R2 value.

Author Response

Responses to reviewer’s comments

Dear Editor and Reviewers,

Firstly I must say my thanks to the reviewers for their support and advises for manuscript improvement.

We carefully read the comments and revised the manuscripts according to the comments.

The comments of the three reviewers and our revision explanations are attached, please find it.

Best regards,

Ke-Shi Zhang (corresponding author)

Reviewer 2 Report

This paper mostly shows a well-structured example of a scientific article, although some word selection must be improved and more iterative results must be inserted to strengthen the argumentation. It is hard to decide that this paper can be considered for revision or complete rejection, as the authors failed to convince their ideas into the manuscript and turn out to be doubtful. This paper area, however, is still in line within the journal scope, and of course, it should follow the high qualification standard of the journal.

Please see attached comment for detail.

Author Response

(The authors gave the same response as above.)

Reviewer 3 Report

1.       A very modest literature review. There are only a few examples of damage accumulation models. There is no references to the type of analysis carried out in the article. But the authors themselves cite examples of similar research, e.g. [30] or [32], later in the paper. Please complete the introduction.

2.       Does this method mean that you have to destroy a specimen to find out what durability the specimen would have? Please explain this doubt in the Introduction.

3.       Please correct some mistakes marked-up in the attached paper.

Author Response

(The authors gave the same response as above.)

Round 2

Reviewer 1 Report

Thank you for all the changes and corrections, in the opinion of this Reviewer, the manuscript is now ready for publication.

However, the final decision belongs to the Editor of course.

Author Response

Responses to reviewer’s comments

Firstly I must say my thanks again to the reviewer for his works for our manuscript improvement.

Best regards,

Ke-Shi Zhang (corresponding author)

Reviewer 2 Report

The authors have made some efforts for the revised version of the manuscript.

However, the terminology used is not appropriate. For example, crack definition is separation or damage development due to the displacement discontinuity surfaces within the solid. Displacement is related to the dislocation of atomic structure. It is doubtful to say that those tiny scratches are cracks. We understand that these marks will appear due to the applied strain on the material, such mechanisms should be clearly discussed on the manuscript. I requested the authors to provide the initial microstructure and after pre-cycling of the investigated steel in advance is to see what the nature of cracking behaviour will be. Those tiny scratches did not act as crack initiation, it is evidenced by the fractography observation showing dimple feature (that the crack was developed from the void in the interior and expanded under continued deformation until separation/fracture), as seen from figs 5&6.

I suggest the authors to re-phrase the word "cracks" by definition in this manuscript.

The method on how to determine the residual strength should be added into the manuscript.

Can you show me similar work as damage characterisation here? Meaning, can this method be used for other materials and application? Since the authors mentioned stainless steel for comparison in author's note. Otherwise, the manuscript must be specific on this study/application, for example the title should be changed with type of pre-fatigue and HRB335 steel, so on.

Author Response

Firstly I must say my thanks again to the reviewer for his works for our manuscript improvement.

We carefully read the comments and revised the manuscripts according to the comments.

The comments of the reviewer and our revision explanations are attached as an attachment, please find it.

Best regards,

Ke-Shi Zhang (corresponding author)
